# The Potential Role of Microorganisms on Enteric Nervous System Development and Disease

**DOI:** 10.3390/biom13030447

**Published:** 2023-02-27

**Authors:** Atchariya Chanpong, Osvaldo Borrelli, Nikhil Thapar

**Affiliations:** 1Division of Gastroenterology and Hepatology, Department of Pediatrics, Faculty of Medicine, Prince of Songkla University, Songkhla 90110, Thailand; 2Neurogastroenterology & Motility Unit, Gastroenterology Department, Great Ormond Street Hospital for Children, London WC1N 3JH, UK; 3Stem Cells and Regenerative Medicine, UCL Great Ormond Street Institute of Child Health, London WC1N 1EH, UK; 4Gastroenterology, Hepatology and Liver Transplant, Queensland Children’s Hospital, Brisbane, QLD 4101, Australia; 5School of Medicine, University of Queensland, Brisbane, QLD 4006, Australia; 6Woolworths Centre for Child Nutrition Research, Queensland University of Technology, Brisbane, QLD 4101, Australia

**Keywords:** enteric nervous system, development, enteric neuropathy, dysmotility, microbiota, microbes, bacteria, viruses, parasites

## Abstract

The enteric nervous system (ENS), the inherent nervous system of the gastrointestinal (GI) tract is a vast nervous system that controls key GI functions, including motility. It functions at a critical interface between the gut luminal contents, including the diverse population of microorganisms deemed the microbiota, as well as the autonomic and central nervous systems. Critical development of this axis of interaction, a key determinant of human health and disease, appears to occur most significantly during early life and childhood, from the pre-natal through to the post-natal period. These factors that enable the ENS to function as a master regulator also make it vulnerable to damage and, in turn, a number of GI motility disorders. Increasing attention is now being paid to the potential of disruption of the microbiota and pathogenic microorganisms in the potential aetiopathogeneis of GI motility disorders in children. This article explores the evidence regarding the relationship between the development and integrity of the ENS and the potential for such factors, notably dysbiosis and pathogenic bacteria, viruses and parasites, to impact upon them in early life.

## 1. Introduction

The enteric nervous system (ENS) is part of the peripheral nervous system that intrinsically innervates the gastrointestinal (GI) tract. It is a vast, mesh-like network of neurons and glia embedded within the bowel wall, extending along the length of the GI tract [1]. It functions as a master regulator and coordinates many of the essential functions of the GI tract, including motility, sensation and secretion [2]. The ENS consists of motor and sensory neurons that innervate muscular layers, secretory glands, and the lymphatic vascular system within the gut wall. These neurons are organized into two plexuses of enteric ganglia and glial cells: the myenteric plexus (Auerbach plexus) and submucosal plexus (Meissner plexus). The ENS forms a sensory-motor reflex circuit composed of intrinsic primary afferent neurons (IPANs), interneurons, and motor neurons (Figure 1A) [3]. Apart from regulating intestinal motility (contraction and relaxation; Figure 1B), the ENS has various roles which include gut-barrier function, sensing of luminal contents and interaction with gut microbiota and its constituent microorganisms including bacteria, viruses and fungi [4]. Physiologically, the GI tract and the ENS maintain a symbiotic relationship with the microbiota [5]. 

As opposed to the central nervous system and, indeed, other parts of the peripheral nervous system of the human body, the proximity to pathogens places the ENS at higher potential risk of damage. Disturbances of the ENS have direct implications for the development of GI-motor and sensory disorders, the so-called GI motility disorders. These may result from complete or selective (e.g., specific-neuronal-subtype-loss) disruption of the ENS in a particular segment or more diffusely. Gut epithelial cells, including tight-junction protein complexes, play a critical role in maintaining intestinal barrier integrity, to allow the transport of essential molecules and restrict harmful external stimuli [6]. ENS disruption can also result in impairment of the intestinal barrier, leading to translocation of microbes or their products. A “leaky gut” could potentially lead to a series of changes involved in the pathophysiology of several diseases, such as inflammatory bowel disease, autoimmune diseases, metabolic disorders and neurodegenerative diseases (e.g., Parkinson’s disease) [6,7,8].

The role of pathogens in GI motility disorders has been more explored in adult patients, with relatively little known about their role in the etiopathogenesis of ENS disorders in children. This seems especially relevant given the potential for dysbiosis and heightened exposure to a number of potentially neuropathic microorganisms in early life (both pre-natal and post-natal) as well as, most importantly, the vulnerability of the ENS and the gut–brain axis during a key period of its development. With reference to available experimental and clinical evidence, this article explores and hypothesises about the potential role of various pathogens in the aetiopathogenesis of paediatric GI motility disorders.

## 2. ENS Development in Prenatal, Postnatal and Childhood Periods

Although complete colonisation of the GI tract by ENS precursors occurs within the first trimester, structural and functional maturation continues until birth and well into the postnatal period [9,10,11,12]. Recent studies have shown that the ENS is critical not only for effecting gut motility but for intestinal-barrier, immune and mucosal-enteric-glial function as well [9,10]. 

During embryogenesis in the mouse [13], enteric neural crest cells, precursors of the ENS, enter the foregut around embryonic day 9.5 (E9.5) with complete colonisation of the developing GI tract by E14.5, followed by a rostrocaudal gradient of organisation of nerve plexuses and ongoing neuronal and glial differentiation. Whilst myogenic contractions are noted in the duodenum by E13.5, neuronal-mediated contractions become present towards the end of the embryogenesis period (E18.5). During early postnatal life, neuronal networks and synaptic transmission in the duodenum become more mature, with neurally mediated motility observed in the colon at later stages. As weaning is achieved, there is further development of neuronal networks, synaptogenesis, neurogenesis, and maturation of myenteric neurons (in mice after day 21 of life until adulthood) Ref. [12]. Kulkarni et al. demonstrated that the myenteric ganglia in adult mice were capable of maintaining neuronal numbers despite evidence of ongoing neuronal loss [14]. These data suggest that the ENS in childhood and adulthood may be capable of repairing itself and perhaps continues to develop throughout life.

With regards to ENS development in humans, Wallace and Burns demonstrated that neural crest cells enter the foregut at embryonic week (EW) 4, and complete colonisation of the terminal hindgut by EW 7 [15]. The foetal intestine shows a mature appearance of both submucosal and myenteric ganglia, muscular layers, and Interstitial cells of Cajal networks by EW 14 [15]. McCann et al. analysed human foetal intestine to elucidate the development of neuronal diversity, network formation and functional maturation in the human ENS [11]. They showed that human foetal colon expressed excitatory neurotransmitter and synaptic markers at the level of the myenteric plexus by foetal EW 12. Following that, nNOS neurons were first detectable at EW 14, followed by the presence of calcium-compound activation at EW 16 [11]. 

## 3. The Microbiota and ENS Development and Disease

For more than a century, many accepted the sterile womb paradigm, believing that foetal development occurred in a sterile state, given investigations failed to identify any microbes in the foetal meconium, amniotic fluid and placenta from healthy pregnancies [16,17,18]. Recently, evidence has emerged indicating that in normal pregnancies, the amnionic fluid and placenta contain non-pathogenic commensal microorganisms [19,20,21,22,23,24]. Additionally, studies have also detected microbial components in umbilical-cord blood, meconium and foetal membranes [25,26,27]. Aagaard et al. proposed placental colonisation through a hematogenous transmission as a possible method for the foetal colonisation before birth [19]. A study in a mouse model showed that maternal microbiota shapes the immune development of their offspring [28]. The pre-natal microbiome could potentially modulate the development and organisation of the ENS [4,29], although this remains to be proven.

Although intestinal microbes and their interaction with the gut have existed for time immemorial, their roles in health and diseases have only truly been elucidated in recent decades [30,31]. The establishment of the microbiota–gut–brain axis appears critical for health, and its disruption is increasingly implicated in disease. It is well established that, at birth, the intestine is rapidly colonised by microbes from their immediate environment, i.e., their mother’s vagina or skin, depending on the mode of delivery [32]. Further key influences on the developing microbiome come from the diet and exposure to the environment [33]. Several studies have reported that human breast milk microbiota and the method of feeding (e.g., breast suckling, bottle feeding or tube feeding) can considerably influence not only the composition of the future microbiota, but also an infant’s health by promoting intestinal immune homeostasis and facilitating digestive processes [34,35,36,37]. Eventually, in most individuals a ‘normal’ microbiota is established, which appears to confer optimal development and health on infants and children. Dysbiosis or microbial imbalance occurs, in theory, where there is gain or loss in the gut microbial community or changes in the relative abundance of microbes (e.g., alteration in the relative proportions of benevolent microbes to pathogenic ones). 

Studies have shown that postnatal ENS development is critically modulated by intestinal microbiota Refs. [12,29,32]. Studies of germ-free mice, with a complete absence of gut microbiota, have revealed structural abnormalities in the ENS as compared to specific pathogen-free mice [12,38,39]. These included a reduction in neuronal density, an increase in the proportion of nitrergic inhibitory neurons, a decrease in serotonin levels and the disruption of the myenteric plexus. Likewise, intestinal motility was also reduced in terms of the frequency and amplitude of contractions, resulting in slow intestinal transit [12,38]. Similarly, postnatal mice exposed to broad-spectrum antibiotics showed disruption of myenteric neurons and glia, a reduction in serotonin levels, and abnormal intestinal transit [12]. Kabouridis et al. also compared the development of enteric glial cells (EGCs) in germ-free and conventionalized mice. They found that the organisation of EGCs and their ongoing supply to the lamina propria were regulated by the gut microbiota [40].

Much of the attention has focused on the role of postnatal dysbiosis as a determinant of disease, including of the ENS. This is not surprising, given our current knowledge that the microbiome is only established at birth. Saying that there is evidence that a ‘pre-natal’ microbiome may exist that could potentially impact the developing foetus. 

Excessive hygiene conditions and antibiotic use during the early stage of life have been shown to compromise the establishment of normal gut microbiota in animal models [41,42]. A recent systematic review of 24 eligible studies concluded that infants from mothers exposed to antibiotics had a decreased microbial diversity (a decreased abundance of Bacteriodetes and Bifidobacteria and increased Proteobacteria) compared to non-exposed infants [43]. Conversely, maternal dental, intestinal and intrauterine (chorioamnionitis) infections could contribute to hematogenous spreading to the placenta and systemic inflammatory responses, which are associated with an increased risk of adverse pregnancy and neonatal outcomes including spontaneous preterm labour, low birthweight, and necrotizing enterocolitis (NEC) [44,45,46]. NEC is increasingly considered a condition related to dysbiosis. Intrauterine exposure of the foetus to chorioamnionitis is strongly associated with an increased risk of NEC in the newborn, particularly preterm infants [47]. In the neonatal intestine the relative immaturity of the intestinal barrier and abnormalities in intestinal permeability may be responsible for augmenting intestinal inflammation, causing NEC [48].

As mentioned earlier, the ENS plays a key role in maintaining the intestinal epithelial integrity and regulation of tight-junction expression [4]. Studies in human and animal models have found ENS abnormalities in the neonates born to maternal chorioamnionitis and those with NEC [49,50,51,52]. In 1998, using immunohistochemistry staining, Sigge et al. evaluated eight intestinal specimens resected from patients with NEC and compared these to tissue from three controls. The authors observed a loss of both glial and neuronal cells in the myenteric plexus. However, the most severe neuronal damage was detected in the mucosal and submucosal layers, with an absence of vasoactive-intestinal-peptide (VIP) and nitric-oxide-synthase (NOS) immunoreactivity in the submucosal plexus [50,51].

Following this, Zhou et al. evaluated eighteen intestinal segments from patients with NEC, compared to eight controls, and fourteen NEC and six control samples at time of stoma closure. They found profound ENS damage, with neuronal apoptosis and reduction of NOS expression in the myenteric plexus of the intestines affected by NEC, compared to the control intestines. These structural abnormalities persisted to the time of stoma closure (2–3 months later) [52]. Similar abnormalities were identified in animal models [49,52]. Whether severe disorders of GI motility and disruption of the ENS are a consequence of NEC has not been confirmed although it remains conceivable that, in some children, severe motility disorders such as paediatric intestinal pseudo-obstruction (PIPO) may be a consequence of dysbiosis, including that leading to NEC, with ultimate disruption of the ENS.

Apart from NEC, alterations in gut microbiota and the ENS in early life have been implicated in the pathogenesis of several diseases. For example, the pathophysiology of inflammatory bowel disease (IBD) appears, in part, to be related to leaky gut syndrome and dysbiosis [53]. Additionally, the number of enteric glia cells is depleted, and their network is disrupted in the intestine affected by IBD [32,54]. Studies have shown that gut microbiota may also play key roles in neurological diseases including neurodevelopmental disorders (e.g., autistic spectrum disorder), and neurodegenerative disorders (e.g., Parkinson’s disease) [12,32].

## 4. Pathogens and ENS Development and Disease

Other than dysbiosis a number of pathogens are potentially capable of affecting the foetus or infant in pre- and postnatal life (Figure 2; Ref. [55]). These include a number of neuropathic pathogens that could specifically affect the ENS, a number of which will be discussed in more detail below. Although much remains to be known about the mechanisms of how microorganisms modulate ENS development, many signaling mechanisms involving direct and indirect pathways have been proposed [5,12]. These include those mediated by toll-like receptors expressed by epithelial and immune cells that have modulatory effects on enteric neurons, glia and neuromuscular function [56], the release of microvesicles, transcription factors, neurotrophic factors and toxins from microbes [57,58], the production of microbial metabolites and indirect communication through enteroendocrine cells and the immune system [12,29].

### 4.1. Bacteria

Many bacteria are capable of causing GI infection and the direct disturbance of ENS function and GI motility both during the acute phase of the illness as well as more chronically thereafter. Postinfectious disorders of GI sensory and motor function are well described [59,60]. Saps et al. studied a cohort of children with acute bacterial gastroenteritis including *Salmonella*, *Campylobacter*, and *Shigella* spp. They reported that approximately one-third of afflicted patients developed functional abdominal pain, including irritable bowel syndrome (IBS; both diarrhoea- and constipation-predominant) and functional dyspepsia following the acute infectious episode [60]. In a seminal follow-up study of a cohort of adult and children following an outbreak of salmonella gastroenteritis, Cremon et al. showed that children, as opposed to adults, were significantly more likely to develop IBS following salmonella infection, suggesting an inherent vulnerability of the ENS, or perhaps the gut–brain axis, to early-life programming [59]. 

It is possible that disturbances in GI motility may also reflect indirect effects on the ENS following bacterial infection. Di Nardo et al. [61] showed that low-grade inflammation and immune activation are often present in children with IBS, manifest as increased numbers of mast cells and apposition to nerve fibres [61]. A study in a mouse IBS model revealed abnormal colonic contractility and reduced response to carbachol in the proximal colon segments of IBS mice compared to controls. Colonic dysfunction was related to abnormally high levels of short-chain fatty acids, which could be due to intestinal dysbiosis in the affected mice [62]. Ostertag et al. [63] analysed neuronal activity in submucosal ganglia that presented in the colonic-mucosal biopsies from IBS patients. The responses to nicotine and electrical nerve stimulation in submucous ganglia was no different between control and IBS patients. However, the biopsies from IBS patients responding to application of the IBS-cocktail (containing serotonin, histamine, tryptase, and TNF-α) was significantly lower, compared to the control samples. This appeared to be due to a lower evoked calcium increase and reduced percentage of responding ganglionic area [63].

A study in a sheep model for pregnancy affected by chorioamnionitis showed that there was a loss of mature neurons and glial cells in the foetal terminal ileum after prolonged exposure to *Ureaplasma parvum* infection in utero [64]. Heymans et al. reported significant ENS injury in the myenteric rather than submucosal plexus of the ovine foetuses exposed to intrauterine infection with *Ureaplasma parvum* [65]. The ENS changes occurred as early as 4 days after the exposure to the intrauterine infection. Although in this experimental model the ENS changes appeared to recover after 15 days of the exposure, it remains to be seen whether motility function was normal and whether this model of ENS damage is relevant to humans. Additionally, premature alterations in the composition of microorganisms in the intrauterine environment can induce an inflammatory response, causing spontaneous miscarriage or initiating preterm labour and delivery [66,67,68]. Microorganisms translocating to the placenta potentially include those from vaginal contamination, oral and bloodstream transmission, and uterine commensal microbiome [19,69,70,71,72]. Contini et al. identified silent infections by detecting deoxyribonucleic acid or DNAs of *Ureaplasma* spp., *Mycoplasma* spp. and *Chlamydia trachomatis* in chorionic villous tissues and peripheral blood mononuclear cells from women with spontaneous abortion [51]. Pathogens such as *Ureaplasma* that cause chorioamnionitis have been implicated in miscarriage and can also affect the ENS in the offspring [64].

### 4.2. Viruses

#### 4.2.1. *Herpes simplex* Virus

It is widely known that *Herpes simplex* virus (HSV) type 1 and 2 infect the epithelial cells of the oro-genital mucosa and then spread to the sensory peripheral nervous system before establishing a life-long latent infection within dorsal root ganglia [73]. In 2010, Brun et al. highlighted the fact that HSV-1 could also cause persistent infection in rat myenteric ganglia after 1–10 weeks of intragastric inoculation [74]. In vitro contractility studies were performed and demonstrated functional changes in the ileum including increased responses to carbachol and CaCl2, resulting in alterations of gastrointestinal transit [74].

Subsequently, the same group developed a murine model of HSV-1 infection of the ENS to understand how this causes intestinal dysmotility. They found that the virus triggers enteric neurons to recruit macrophages via the production of specific chemoattractant factors (Toll-like receptor 2 and monocyte chemoattractant protein-1), inducing an inflammatory reaction responsible for gut dysfunction [75,76].

HSV-1 infection appears capable of causing intestinal dysmotility anywhere in the GI tract. A study in 11 Japanese patients with achalasia (age range 27–78 years) identified HSV-1-encoded microRNAs in biopsy samples of the lower-oesophageal-sphincter (LOS) muscle during per-oral endoscopic myotomy. They concluded that the muscular layer of the LOS may serve as an HSV-1 reservoir, given the expression of viral microRNA, and that HSV-1 infection could contribute to disease pathogenesis in achalasia [77]. Becker et al. comprehensively analysed epidemiological and genotype–phenotype data from 696 European adults with achalasia and reported that the most prevalent cause of achalasia was related to viral infections (particularly from the varicella-zoster virus) [78]. However, there are scarce data in paediatrics, and similar studies in children with achalasia are warranted. 

Studies in mice with HSV-1 infection showed that mice often died due to toxic megacolon. Chaudhury and colleagues postulated that HSV particles may affect and cause smooth-muscle myopathy by acting on sildenafil response proteins, leading to delayed intestinal transit [79]. Conversely, Khoury-Hanold suggested that direct injury and damage of the enteric neurons by HSV-1 that spreads via the dorsal root ganglia to the ENS in the colon caused permanent loss of peristalsis and the development of toxic megacolon [80]. This phenomenon was reversed by treating the mice with polyethylene glycol [79,80].

#### 4.2.2. *Varicella zoster* Virus

*Varicella zoster* virus [VZV) or *Human herpesvirus 3* behaves similarly to HSV. After primary infection and during viremia, it becomes latent within neurons of the cranial nerve and dorsal root ganglia, as well as in sensory, autonomic neurons and the ENS [81]. Chen et al. identified VZV DNA and transcripts in 100% (6/6) of resected bowel from children with a history of varicella infection and 85% (6/7) from children who had had the varicella vaccine [82]. Likewise, Gershon et al. found VZV in at least some ganglia of seven children at autopsy (six were known to have been vaccinated), despite the lack of any record of clinical varicella. Additionally, VZV DNA and RNA were identified in trigeminal, cervical, thoracic, and lumbar ganglia from the studied patients [83]. Both studies [82,83] also looked into the pathophysiology of latent VZV infection in the nerve ganglia using animal models, and proposed that retrograde transport from infected skin and viremia may deliver VZV to neurons in which the virus becomes latent [82,83]. For example, neurons in the dorsal root ganglia of guinea pigs appeared to project to both the skin and the intraperitoneal viscera [82]. Thus, VZV reactivation in the enteric neurons or any extrinsic neurons that project to the bowel can cause ‘enteric zoster’, characterised by a gastrointestinal disorder without cutaneous manifestations [81]. 

VZV infection is known to be associated with intestinal pseudo-obstruction or Ogilvie’s syndrome in adult cases [84,85]. A few case reports in children have demonstrated the association between VZV infection and GI neuromuscular disorders. Windster et al. described a case of 7-month-old infant presenting with signs and symptoms of ileocolic intussusception before developing VZV skin manifestations. VZV DNA was not only detected from the swab on the skin lesions, but its RNA was also found in the myenteric plexus and submucosal nerve fibres of the resected intestine. Although there were no signs of abnormalities in the myenteric ganglia, recruitment of lymphocytes causing enlarged lymph nodes was noted, which could provide a lead point for intussusception [86].

Another case report described a 6-year-old boy presenting with the classic features of congenital varicella syndrome at birth, including growth restriction, microphthalmia, Horner syndrome, hypoplastic left arm, hypotonia and micropenis. He also had signs and symptoms of gut obstruction at birth, which was subsequently known to be from colonic atresia. Although VZV serology obtained from the infant was negative, the diagnosis was based on maternal positive-VZV serology during pregnancy and the presence of cutaneous herpes zoster at 3 months of age [87]. Colonic atresia in congenital VZV infection is thought to be from in utero ischemia and necrosis from volvulus or interruption of the mesenteric blood supply. Moreover, the viral injury to the enteric plexuses may also lead to direct ENS damage and/or contribute to poor blood supply, leading to intestinal atresia [87].

#### 4.2.3. *Cytomegalovirus*

Human *cytomegalovirus* (CMV) can be found in nearly all body secretions, including saliva, urine, breast milk, cervical and vaginal secretions, semen, and blood. Maternal CMV infection can lead to either congenital or perinatal infection in offspring [88]. GI manifestations of CMV infection are not common, especially in immunocompetent hosts. A case series reported on three clinical cases where CMV infection of the foetus or placenta was associated with antenatal paralytic/meconium ileus [89]. Their autopsies showed no chorioamnionitis, but profound placentitis, with evidence of CMV infection (typical nuclear inclusions). Of these three cases, two had positive-CMV cells in the neurons of the GI tract [89]. There have also been a few case reports and series associating neonatal surgical conditions with CMV infection [90,91,92]. 

Bonnard et al. reported a series of five neonates presenting with GI conditions that required surgical operation [90]. All patients had surgical specimens showing typical CMV nuclear inclusions. These conditions included necrotising enterocolitis, volvulus of Meckel’s diverticulum, perforation of Meckel’s diverticulum and distal ileal atresia [90]. Yeung et al. recently described a late preterm infant having clinical presentations that mimicked NEC, subsequently requiring laparotomy in view of intestinal obstruction. A segment of distal colonic stricture was found and resected; mixed inflammation and CMV intranuclear inclusion bodies were detected on histopathology [91]. Another case reported coexistent congenital-CMV infection with intestinal malrotation in a term infant who was prenatally suspected of small-bowel dilatation [92]. Among these cases, all had positive intestinal-CMV biopsies. Of note, in many cases the virus was also demonstrable in the intestinal neurons. No information was available on specific ENS abnormalities or the integrity of GI motility. 

A specific rare but severe GI motility disorder characterised by a clinical picture of intestinal obstruction but in the absence of luminal occlusion, is paediatric intestinal pseudo-obstruction (PIPO) [93,94]. The majority of PIPO cases exhibit neuropathic rather than myopathic disturbance, suggesting that the ENS is a key target [94]. In the adult counterpart of PIPO, chronic intestinal pseudo-obstruction (CIPO) there has been considerable and recent focus on the role of viral infections, with some experts proposing that a search for potential viral infections should be inherent to investigations in paediatric and adult cases [95]. The most common neuropathic viruses include human polyomavirus or JC virus, HSV, CMV, VZV, the Epstein–Barr virus (EBV), and Flaviviruses; several reports have revealed the DNA of these viruses within glial and neuronal cells of the myenteric plexus in the GI tract of patients with intestinal pseudo-obstruction [95,96,97].

Most, if not all, of children with PIPO receive multiple surgical interventions, unfortunately many of these prior to a definitive diagnosis [98,99]. To date, the detection of viral particles in GI specimens is rarely described in PIPO patients. To our knowledge, there are four case reports describing infants presenting with symptoms that mimic intestinal obstruction [88,100,101,102]. All patients underwent surgical interventions that showed no evidence of malrotation or organic obstruction, but their intestinal specimens revealed abnormalities in enteric nerves (e.g., hypoganglionosis, thickening of nerve processes) and intranuclear inclusions in enteric neurons. Subsequently, CMV infection was also confirmed by viruria and specific IgM and IgG antibodies. It is unclear whether CMV is the direct cause for the pseudo-obstruction or an opportunistic superinfection in the presence of pre-existing inflammation or dysmotility.

#### 4.2.4. *Rotavirus*

*Rotaviruses* are non-enveloped double-stranded ribonucleic acid (dsRNA) viruses that have a complex architecture; the RNA segments encode six structural viral proteins (VP1, VP2, VP3, VP4, VP6 and VP7) and six non-structural proteins (NSP1, NSP2, NSP3, NSP4, NSP5 and NSP6) [103]. The virus is transmitted primarily through the faecal–oral route. The virus causes infection by attaching to host cells using the outer capsid protein VP4, then replicates in the mature enterocytes located in the middle and tip of the villi [103]. Although the common presentation is secretory diarrhoea, the virus is also able to affect the ENS and modulate intestinal motility through NSP4 (a viral enterotoxin secreting from rotavirus-infected cells) [104]. 

The *rotavirus* infection and NSP4 enterotoxin signal through phospholipase C to increase intracellular calcium levels, which subsequently induce the secretion of serotonin (5-HT) from enteroendocrine cells in both humans and mice. Serotonin can activate enteric neurons and is involved in the regulation of intestinal motility and secretion, as well as blood flow [105,106]. Recently, Hellysaz et al. revealed that the *rotavirus* infection was able to disrupt the autonomic nervous system by downregulating noradrenergic sympathetic nerves in the ileum, which resulted in more-rapid intestinal transit in mice [107]. 

Although it is known that the *rotavirus* infection increases intestinal motility, children with rotaviral gastroenteritis usually suffer from nausea and vomiting. Bardhan et al. performed liquid-gastric-emptying scintigraphy in 10 children, and reported that the infection was accompanied by abnormal gastric motor function or delayed gastric emptying [108]. This disturbance in motility can also be seen after the acute infection akin to a post-viral gastroparesis [109]. A case report also showed that rotavirus gastroenteritis could be complicated by toxic megacolon [110]. Several mechanisms have been proposed for toxic megacolon including direct damage of the intestinal muscle, electrolyte disturbance, and changes in colonic response to intestinal inflammatory mediators such as nitric oxide (NO). Interestingly, a study in an animal model demonstrated that mRNA expression of NOS in the ileum was upregulated in rotavirus-infected mice, specifically during the course of diarrhoea [111]. NO is a neurotransmitter released from inhibitory neurons to relax the intestinal muscle, which in turn may lead to gut dilatation [2]. Recent evidence suggested the rotavirus infection could also alter the gut-microbiota composition in suckling mice [112].

#### 4.2.5. *Coronavirus SARS-CoV-2*

Since 2019, the novel *coronavirus SARS-CoV-2* has become a worldwide issue. It enters host cells by binding the viral spike with the angiotensin-converting enzyme-2 (ACE-2) receptor, which is highly expressed on pulmonary alveolar epithelial cells. The ACE-2 receptor is also expressed in extrapulmonary tissues, including the GI tract, heart, liver, and kidney [113]. After binding to the ACE-2 receptor, virus entry requires cleavage of the spike protein by a type II transmembrane serine protease (TMPRSS2). Thus, both ACE-2 and TMPRSS2 are necessary for viral entry into cells [114].

In the GI tract, it has been reported that *SARS-CoV-2* could possibly replicate through the ACE-2 receptor, which is also highly expressed on intestinal enterocytes [115,116,117]. Following viral replication, local inflammation may impact and impair the mucosal barrier leading to epithelial damage and mucosal inflammation, which in turn may contribute to the development of GI symptoms in COVID-19 patients Ref. [113].

Deffner et al. [118] found that ACE-2 and TMPRSS2 are also expressed by enteric neurons and glial cells, both in the myenteric and submucosal plexuses of the small and large intestine. Since the ENS controls all GI function including intestinal motility, secretion, blood flow, epithelial barrier and immune system [4], its dysfunction could contribute to different GI manifestations, including but not limited to, diarrhoea, nausea, vomiting, dysphagia, and abdominal pain [117,118,119]. More recently, Furuzawa-Carballeda and colleagues discovered SARS-CoV-2 and its receptor in the lower-oesophageal-sphincter (LOS) muscle complex of achalasia patients who had COVID-19, but not in achalasia patients without COVID-19 or controls. This study postulated that the presence of the *SARS-CoV-2* virus in the LOS may be associated with the development of post-COVID-19 achalasia [120]. Furthermore, the association between SARS-CoV-2 infection and ileocolic intussusception has been postulated; a recent report revealed the presence of the virus in the mesenteric lymph node and the ileum of the infected children [121].

A retrospective study based on a library of second-trimester human foetal tissues reported that although the rate of vertical transmission was low, the GI tract was more likely to be susceptible to infection, since it is exposed to potentially infected amniotic fluid. In addition, high susceptibility of COVID-19 could be due to the high expression of both ACE-2 and TMPRSS2 in the GI tract [122]. COVID-19 may also alter intestinal microbiota, leading to dysbiosis, since ACE2 plays a key role in gut-microbiota regulation [121]. Apart from the direct effects of the virus on dysbiosis, changes in the maternal–foetal environment, adaptations in perinatal care (e.g., higher rate of caesarean section, decreased breast-milk usage) and changes in hygiene, due to the pandemic, may also have a role to play in altering the microbiota composition of offspring [123].

#### 4.2.6. *Bacteriophages*

Other pathogens with the potential to affect the ENS, although through neurodegeneration, are bacteriophages. These bacterial viruses have been shown to modulate gut permeability and lead to leaky gut syndrome in animal models [124]. Tetz et al. recently reported an alteration of gut-phagobiota composition in patients with Parkinson’s disease [124]. Their effect on the ENS in early life is not known.

### 4.3. Parasites

In a similar way to bacterial and viral infection, parasitic invasion in the gut disturbs the balance between the host and the gut microbiota and induces immune responses to recognise and eliminate the pathogen. Helminth infection triggers type 2 immune responses by releasing interleukin (IL)-4, IL-5, and IL-13 from innate lymphoid cells type 2 (ILC2) to CD4 + T cells (Th2), which function to accelerate parasite expulsion via the ‘weep and sweep’ response. Thus, parasitic infestation could potentially also alter or modulate the ENS [125,126].

Although some helminths can enhance the gut-barrier function, restrict bacterial translocation and induce a tolerogenic phenotype of resident muscularis macrophages, which are trained guardians of enteric neurons [126,127], virulent parasitic protozoa can have reverse effects, damaging the gut-barrier function and enteric neurons [128]. 

Adult mice infected with *Trypanosoma cruzi* exhibited delayed GI transit time, intestinal muscular inflammation, and reduced numbers of neurons per ganglion in the colonic myenteric plexus [129]. Likewise, chronic infection with Toxoplasma gondii in the rat model induced neuronal cell death and hypertrophy in the remaining submucosal enteric neurons and damaged the colonic mucosa of the infected rats [130]. It remains unclear whether parasites, or, conversely, a lack of parasites, within the GI tract in early life may have effects on the development or function of the ENS and gut motility.

### 4.4. Fungi

Little is known and reported regarding the effect of fungi on ENS development and integrity. Recent research [131] investigated the role of gut fungi on intestinal physiology and systemic immune development by using adult germ-free dams that were colonized with either defined bacteria, or yeast, or a mixture of bacteria and yeasts or kept germ-free. Although the authors found neurally induced ion transport in the ileum mainly depended on bacterial, but not fungal, colonization, they concluded that fungi have a strong influence on microbiome dynamics and may help promote early-life systemic and gut immunity. Nevertheless, further experiments are warranted to evaluate the role of fungi in the ENS development and diseases.

## 5. Conclusions

The ENS in early life presents a significant potential target for pathogens acting directly through infection or via disruption of the microbiota. Evidence of dysbiosis and infection by virulent pathogens, including bacteria, viruses and other organisms, causing paediatric motility is sparse. Data is accumulating which suggests that such factors could feasibly contribute to ENS injury, causing functional abnormalities and overt disease, and that they should be looked for as part of the diagnostic work-up across the range of motility disorders in children. Better access to GI and enteric neural tissue even in young children, improved analysis (PCR, rapid tests, etc.) to detect pathogens as well as enhance the characterisation of associated ENS defects, will be important to better understand the contribution of pathogens to ENS disorders. This may lead to better preventative strategies, improved diagnosis, and more effective therapies for GI motility disorders in children.

## Figures and Tables

**Figure 1 biomolecules-13-00447-f001:**
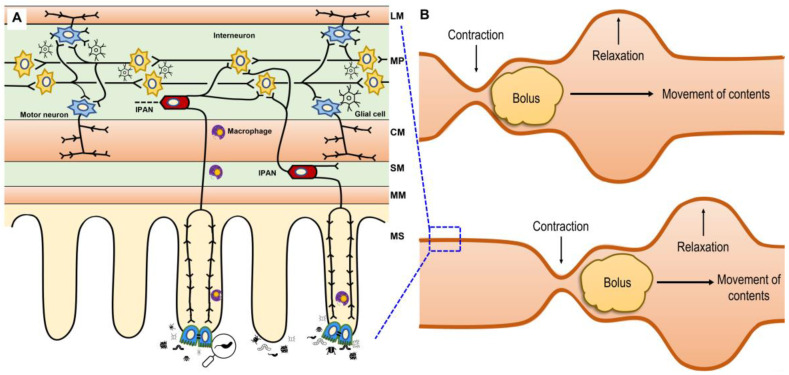
Intestinal peristalsis controlled by the ENS consisting of intrinsic primary afferent neurons (IPANs), interneurons, motor neurons and other supporting cells (e.g., macrophage and glial cell) (**A**); it consists of coordinated contraction and relaxation in different intestinal segments (**B**).

**Figure 2 biomolecules-13-00447-f002:**
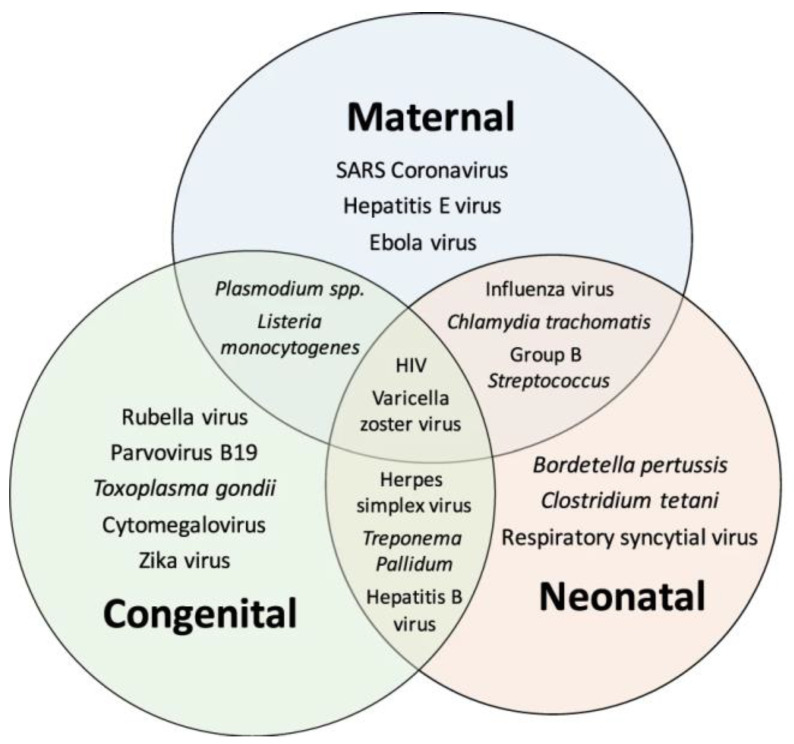
Infectious microbes known to cause maternal complications and affect the infant in pre- and postnatal life [55].

## Data Availability

Not applicable.

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
