# Peer review of "The Potential Role of Microorganisms on Enteric Nervous System Development and Disease"

_biomolecules, 2023, doi:10.3390/biom13030447_

Round 1

Reviewer 1 Report

A very nicely presented review, I have no major comments regarding its contents.

Author Response

Thank you very much for your review of our manuscript and kind comment, which is greatly appreciated.

Reviewer 2 Report

Abstract

the enigmatic microbiome – what do authors mean

Increasing attention is now being paid to the potential of disruption of the microbiome and pathogenic micro-organisms in the potential aetiopathogeneis of GI motility disorders in children. – why “micro-organisms” hyphenated? Please note that microbiome is the totaligy of bacterial genes and microbiota is the totality of bacteria. It seems microbiota is more appropriate here. Please check the manuscript

The description of the ENS is vague – has to be described in more detail, highlighting the role of the intestinal integrity and leaky gut in ENS alterations

Line 44 interaction with the microbiome – please replace with  microbiota

Line 46 – other nervous systems  - what are other nervous systems

Line 48 “motility disorders” – this is unclear

The introduction lacks a clear description of the GI motility disorders

Line 61 - (Reviewed in Ref 4 and 5). – please replace it with a regular citation style

Lines 62-87, figure 1– how is this partis relevant to the manuscript? Has to be removed

Lines 117-127 – have to be the first part of this section to keep the chronology

Part# 3. The microbiome and ENS development and disease  - is vague. The authors suggest that the only disease associated with ENS is NEC, but that is not accurate. This part has to be rewritten. With adding……. The authors need to describe the mechanisms of how different representatives of microbiota modulate ENS development. There are a lot of great articles and reviews on this topic

Name of “4. Other pathogens and ENS development and disease” – what do authors mean under Other pathogens?

Fig 2 has different microorganisms and viruses listed in section #4.

Section 4 4. Other pathogens and ENS development and disease – the mechanisms of how microorganisms are implicated in ENS alterations should be described

Lines 201-208 – are not related to this part

4.2.2. Varicella  - what do authors mean under Varicella? Please correct the name

For the whole of section 4 – remove the description of viral infections which are not relevant to the topic of the article (i.e. ENS development and its alterations).

4.2.5. COVID-19 – it isnot virus. Please double-check the name of the disease vs the virus

Part 4 – what about bacterial viruses (bacteriophages): https://www.nature.com/articles/s41598-018-29173-4

The authors miss the description of the whole plethora of articles linking ENS and neurodegeneration including Braac theory etc.

Author Response

Abstract

the enigmatic microbiome – what do authors mean

‘Enigmatic’ related to the fact that there remains much about the microbiota that remains to be elucidated. We have removed this, corrected the term ‘microbiome’ to ‘microbiota’ and defined ‘microbiota’.

Increasing attention is now being paid to the potential of disruption of the microbiome and pathogenic micro-organisms in the potential aetiopathogeneis of GI motility disorders in children. – why “micro-organisms” hyphenated? Please note that microbiome is the totaligy of bacterial genes and microbiota is the totality of bacteria. It seems microbiota is more appropriate here. Please check the manuscript

Thank you for your comments. We have removed the hyphens and also replaced ‘microbiome’ with ‘microbiota’.

The description of the ENS is vague – has to be described in more detail, highlighting the role of the intestinal integrity and leaky gut in ENS alterations

Thank you. We have described the ENS in more detail and highlighted the role of the intestinal integrity and leaky gut in ENS alterations (line 47-53).

Line 44 interaction with the microbiome – please replace with microbiota

We have amended this as requested.

Line 46 – other nervous systems  - what are other nervous systems

We have changed he sentence to read ‘other parts of the peripheral nervous system’

Line 48 “motility disorders” – this is unclear

We have changed it to ‘GI motility disorders’

The introduction lacks a clear description of the GI motility disorders

We have added a description of GI motility disorders.

Line 61 - (Reviewed in Ref 4 and 5). – please replace it with a regular citation style

We replaced these as suggested.

Lines 62-87, figure 1– how is this partis relevant to the manuscript? Has to be removed

Thank you for the comment. We intended this figure to illustrate the fact that significant processes of ENS development and maturation occur in early life from the period after embryogenesis and prior to birth as well as in early postnatal life. This underlies the vulnerability of the ENS to pathogens and disruption of the microbiota.  

Lines 117-127 – have to be the first part of this section to keep the chronology

Thank you very much, we have moved this paragraph to the beginning of the section.

Part# 3. The microbiome and ENS development and disease  - is vague. The authors suggest that the only disease associated with ENS is NEC, but that is not accurate. This part has to be rewritten. With adding……. The authors need to describe the mechanisms of how different representatives of microbiota modulate ENS development. There are a lot of great articles and reviews on this topic

Thank you. We initially highlighted NEC as one of a number of diseases associated with disruption of microbiota and ENS development. Within the limitations of word count we have now expanded this section to briefly highlight putative mechanisms as well as other diseases associated with ENS disruption.

Name of “4. Other pathogens and ENS development and disease” – what do authors mean under Other pathogens?

Thank you very much. We meant organisms other than microbiota that have the potential to affect ENS development and cause related disease. We have amended the title

Fig 2 has different microorganisms and viruses listed in section #4.

Thank you very much for addressing this. In Fig 2 we set out to highlight that in the perinatal period there are a range of microorganisms that are able to act as pathogens. We sought to highlight those neuropathic pathogens that have been shown to affect the ENS. Others lack evidence of ENS effects and were not elaborated upon.

Section 4 4. Other pathogens and ENS development and disease – the mechanisms of how microorganisms are implicated in ENS alterations should be described

Thank you. We have now added a brief description of the potential mechanisms (line 246-248, 300-303, 324-329, 413-417, 418-421, 443-454, 488-492).

Lines 201-208 – are not related to this part

Thank you very much. We have modified this section (line 247-264).

4.2.2. Varicella  - what do authors mean under Varicella? Please correct the name

Thank you for pointing this error out, which has now been corrected.

For the whole of section 4 – remove the description of viral infections which are not relevant to the topic of the article (i.e. ENS development and its alterations).

Thank you. We have rechecked the section to limit descriptions to those viruses known, in experimental and clinical settings, to affect the ENS.

4.2.5. COVID-19 – it is not virus. Please double-check the name of the disease vs the virus

Apologies, we have corrected this.

Part 4 – what about bacterial viruses (bacteriophages): https://www.nature.com/articles/s41598-018-29173-4

Thank you very much for the suggestion, we have added these in a brief section (4.2.5).

The authors miss the description of the whole plethora of articles linking ENS and neurodegeneration including Braac theory etc.

Our article aimed to focus on early life and effects on the ENS during this time rather than later neurodegeneration. We have now added a brief comment in the introduction (line 69-70) and section 3 (line 195-202).

Reviewer 3 Report

The review deals with the involvement of gut microbiota in morphofunctional and developmental changes of the enteric nervous system with a special focus on the pediatric period. The manuscript is well written. Some changes are suggested to improve the paper (see file attached).

Author Response

The review deals with the involvement of gut microbiota in morphofunctional and developmental changes of the enteric nervous system with a special focus on the pediatric period. The manuscript is well written.

Thank you for your kind comment.

Some changes are suggested to improve the paper (see file attached). 

  1. Consider the use of the term microbiota instead of microbiome. A definition is needed.

Thank you very much. We have made this amendment throughout the manuscript.

  1. Do you mean "innervation". Or is this a neologism for nerve invasion?

Thank you very much. We have removed this figure.

  1. Acronyms must be explained.

Thank you very much. We have removed this figure.

  1. The presence of a specific microbiota in the maternal milk and the different ways of nursing (breast suckling, bottle feeding) can influence the development of the future microbiota. These aspects must be described.

We have added a brief description in section 3 (line 136-140).

  1. Bacterial strains in italics.

Thank you very much. We have amended this.

  1. Virus strains in italics.

Thank you very much. We have amended this.

  1. This acronym is not explained. See lines 225, 256 and 266.

Thank you very much. We have amended this (line 294).

  1. Parasite strains in italics.

Thank you very much. We have amended this.
